# Non-invasive regional cerebral blood flow quantification in the 123I-IMP autoradiography using artificial neural network

**Tetsuro Kaga** [1]*, **Hiroki Kato**[1], **Toyohiro Imai**[2], **Tomohiro Ando**[1], **Yoshifumi Noda**[1], **Takayuki Miura**[2], **Yukiko Enomoto**[3], **Fuminori Hyodo**[4], **Toru Iwama**[3], **Masayuki Matsuo**[1]

1 Department of Radiology, Gifu University, Gifu, Japan, 2 Department of Radiology Services, Gifu University Hospital, Gifu, Japan, 3 Department of Neurosurgery, Gifu University Graduate School of Medicine, Gifu, Japan, 4 Department of Radiology, Frontier Science for Imaging, Gifu University, Gifu, Japan

* kagatetsurow@gmail.com

## Abstract

### Purpose

Regional cerebral blood flow (rCBF) quantification using 123I-N-isopropyl-p-iodoamphetamine (123I-IMP) requires an invasive, one-time-only arterial blood sampling for measuring the 123I-IMP arterial blood radioactivity concentration (Ca10). The purpose of this study was to estimate Ca10 by machine learning (ML) using artificial neural network (ANN) regression analysis and consequently calculating rCBF and cerebral vascular reactivity (CVR) in the dual-table autoradiography (DTARG) method.

### Materials and methods

This retrospective study included 294 patients who underwent rCBF measurements through the 123I-IMP DTARG. In the ML, the objective variable was defined by the measured Ca10, whereas the explanatory variables included 28 numeric parameters, such as patient characteristic values, total injection 123I-IMP radiation dose, cross-calibration factor, and the distribution of 123I-IMP count in the first scan. ML was performed with training (n = 235) and testing (n = 59) sets. Ca10 was estimated in testing set by our proposing model. Alternatively, the estimated Ca10 was also calculated via the conventional method. Subsequently, rCBF and CVR were calculated using estimated Ca10. Pearson's correlation coefficient (r-value) for the goodness of fit and the Bland–Altman analysis for assessing the potential agreement and bias were performed between the measured and estimated values.

### Results

The r-value of Ca10 estimated by our proposed model was higher compared with the conventional method (0.81 and 0.66, respectively). In the Bland–Altman analysis, mean differences of 4.7 (95% limits of agreement (LoA): −18–27) and 4.1 (95% LoA: −35–43) were observed using proposed model and the conventional method, respectively. The r-values of

**Data Availability Statement:** All relevant data are within the paper and its Supporting Information files.

**Funding:** The authors received no specific funding for this work.

**Competing interests:** The authors have declared that no competing interests exist.

rCBF at rest, rCBF after the acetazolamide challenge, and CVR calculated using the Ca10 estimated by our proposed model were 0.83, 0.80 and 0.95, respectively.

## Conclusion

Our proposed ANN-based model could accurately estimate the Ca10, rCBF, and CVR in DTARG. These results would enable non-invasive rCBF quantification in DTARG.

## Background

Regional cerebral blood flow (rCBF) is a useful measure for evaluating cerebral circulation in patients with ischemic cerebrovascular disease and for elucidating the type of dementia [1]. Single-photon emission computed tomography (SPECT) has been employed for the measurement of rCBF using 123I-N-isopropyl-p-iodoamphetamine (123I-IMP), 99mTc-hexamethyl-propyleneamine oxime, and 99mTc-ethylcysteinate dimer as a radioactive tracer. 123I-IMP in particular is an ideal tracer owing to its high first-pass extraction and negligible back diffusion [2]. After the initial uptake in the lung endothelium, 123I-IMP crosses the blood–brain barriers with a high first-pass extraction and demonstrates long retention times in the brain tissue [2, 3]. The initial 123I-IMP SPECT image during the first 10 to 30 min represents good linear correlation with the rCBF distribution, and approximately 8% of the total injected dose accumulates in the brain tissue [4].

Methods for rCBF quantification have been developed during the past few decades. One of the most popular methods for 123I-IMP rCBF quantification is autoradiography (ARG), which is based on a two-compartment model for tracer kinetics [5, 6]. This method requires a one-time-only arterial blood sampling using the standard input function that is previously validated by a population-based collection of data [5–7]. Moreover, the development of the dual-table ARG (DTARG) method can evaluate rCBF at rest and determine cerebral vascular reactivity (CVR) after the administration of acetazolamide in a single-day session [8]. However, DTARG also requires a one-time-only arterial blood sampling that involves the risk of complications and sampling error. Alternatively, Iida *et al*. have proposed a non-invasive method for estimating 123I-IMP arterial blood radioactivity concentration 10 min following the 123I-IMP intravenous injection (Ca10), using body weight and injecting a 123I-IMP dose volume [9]. Although this calculation software is commercially and easily available, the average difference was relatively large [9].

Machine learning (ML) is a branch of artificial intelligence and computer science, which can be described as the process of learning a target function that automatically maps input variables to output variables. Artificial neural network (ANN) is one of the most popular ML algorithms inspired by the function of biological neural networks in the human brain, which mimic the capacity of neurons to learn from past data in the central nervous system [10]. Thus, this study aimed to establish a non-invasive method for estimating Ca10, rCBF and CVR in DTARG using ANN regression analysis.

## Materials and methods

### Participants

This retrospective study was approved by Gifu University Institutional Review Board. The requirement of informed consent was waived due to the retrospective nature of the study. A

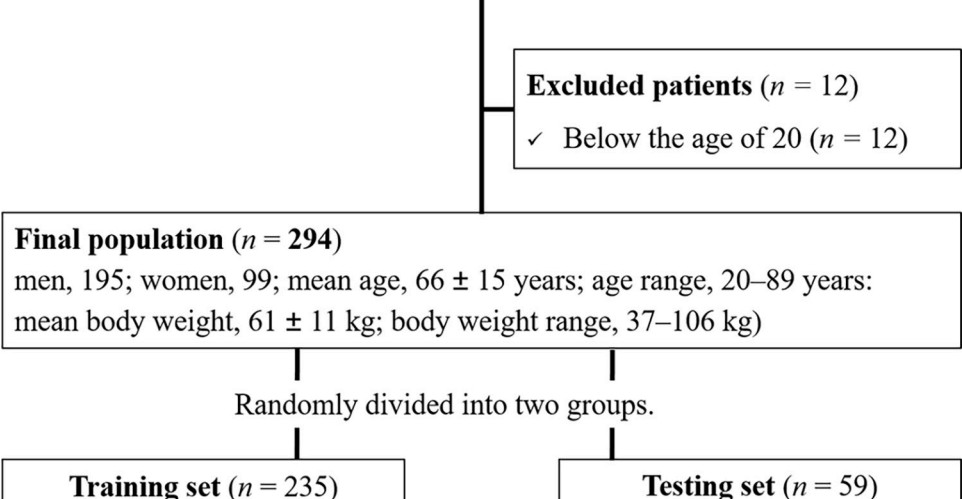

**Fig 1. Participant flowchart.**

series of 306 consecutive participants who underwent DTARG at our institution from January 2016 to December 2020 was included in the subsequent analysis. Overall, 12 of the 306 participants were excluded because they were under 20 years of age. The remaining 294 participants (mean age, 66 ± 15 years; age range, 20–89 years; 195 men; mean body weight, 61 ± 11 kg; body weight range, 37–106 kg) were included in this study (Fig 1). Of these 294 participants, one had normal blood flow, and 293 had abnormal blood flow in DTARG, diagnosed by one radiologist (____with 7 years of post-training experience in nuclear medicine). Furthermore, 176 participants had internal carotid artery stenosis or occlusion (right, 82; left, 54; bilateral, 40), 50 had middle cerebral artery stenosis or occlusion (right, 17; left, 30; bilateral, three), 39 participants had moyamoya disease, seven had vertebral artery stenosis or occlusion (right, three; left, two; bilateral, two), three had basilar artery stenosis, two had subclavian artery stenosis (right, one; bilateral, one), one had bilateral common carotid artery stenosis, one had dual arteriovenous fistula, and 15 participants had multiple cervical and/or intracranial arterial stenosis or occlusion. Of the 294 participants, 54 were examined after treatment for cerebral circulation.

## DTARG imaging protocol

SPECT was performed using a dual-head gamma camera (Infinia™; GE Healthcare, Buckinghamshire, UK) and an ELEGP collimator (GE Healthcare, Buckinghamshire, UK). The energy range was centered at 159 keV with a width of 10%, and a 2-min rotation was performed 14 times in a continuous mode. The matrix size was 64 × 64 pixels.

The procedures of DTARG are presented in Fig 2 [11]. A two-time intravenous injection of 123I-IMP (111 MBq product, respectively) was administered by a radiologist at a 30-min interval. The injected dose volume was calculated by the following equation:

Injected dose volume (MBq) = $111 \times exp(-0.0525 \times t)$

where $t$ represents the elapsed time from the Japan standard time (UTC +9) of 12:00. SPECT data collection commenced simultaneously with the intravenous injection of 123I-IMP. Dynamic SPECT scanning was performed for 28 min (0–28 and 30–58 min after the start of the first injection of 123I-IMP). Approximately 10 min after the first 123I-IMP intravenous injection, 2 mL of arterial blood was obtained from the radial or femoral artery, and the standard input function was calibrated. The sample arterial blood radioactivity concentration was measured using a well counter (DCM-200$^{TM}$; ALOKA, Tokyo, Japan). Approximately 20 min after the first 123I-IMP intravenous injection, 1,000 mg of acetazolamide was intravenously administered. The cross-calibration factor (CCF) between the SPECT images and the well counter system was determined in advance by comparing the average pixel counts derived from the regions of interest (ROIs) on the reconstructed emission images and the well counter radioactivity-counting rate.

The uniformity correction and center-of-rotation calibration were performed using the clinical routine software on the SPECT system. Stereotactic anatomic standardization was also performed on the scan data using three-dimensional stereotactic surface projections (3D-SSP, Nihon Medi-Physics, Tokyo, Japan) [12]. The ROIs were placed on the standardized images both at rest and following the acetazolamide challenge, using an automatic ROI definition software (NEURO FLEXER, Nihon Medi-Physics, Tokyo, Japan) [13]. This software automatically placed three-dimensional ROIs on 10 arterial territories, including the bilateral anterior cerebral artery, bilateral middle cerebral artery (MCA), bilateral anterior part of MCA, bilateral posterior part of MCA, and bilateral posterior cerebral artery. The ROIs were also placed on 10 segments, including the bilateral hemisphere, bilateral basal ganglia, bilateral thalamus, bilateral cerebellar hemisphere, pons, and vermis of the cerebellum (Fig 3).

## Datasets

All clinical and radiological data, including participants' demographics, SPECT data, and the Ca10 of the sample were obtained from the database of the radiology information system. Participants' demographics included sex, age, height, body weight, and injected dose volume. SPECT data from the first scan comprised of CCF, whole radioactivity counts of the head, detector count rate of the head, and radioisotope counts of 10 arterial territories and 10 segments. The Ca10 of the sample was used as an objective variable, and the other 28 items were used as a set of inputs for ANN. The raw datasets are available in S1 Table.

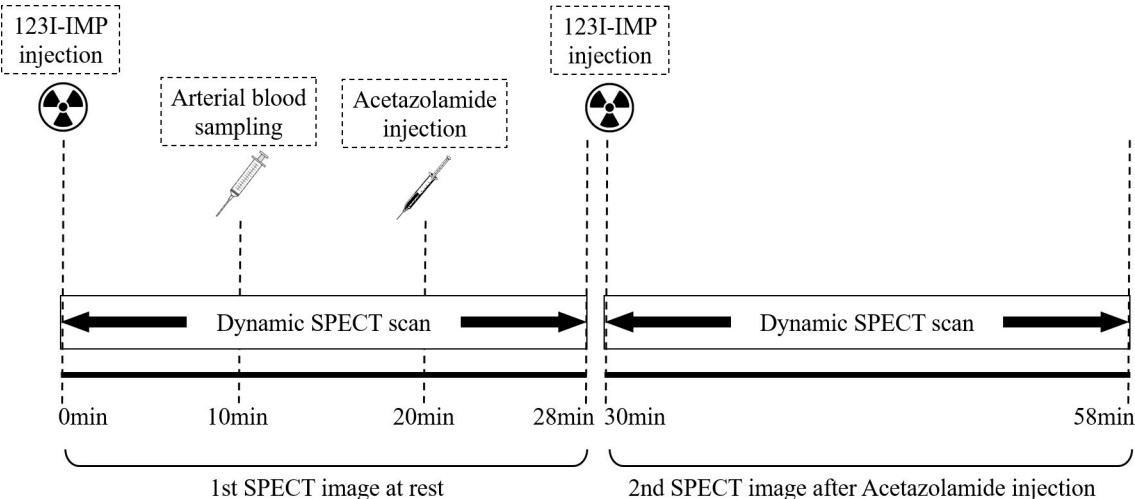

**Fig 2. Dual-table autoradiography procedures.** Note. SPECT, Single-photon emission computed tomography.

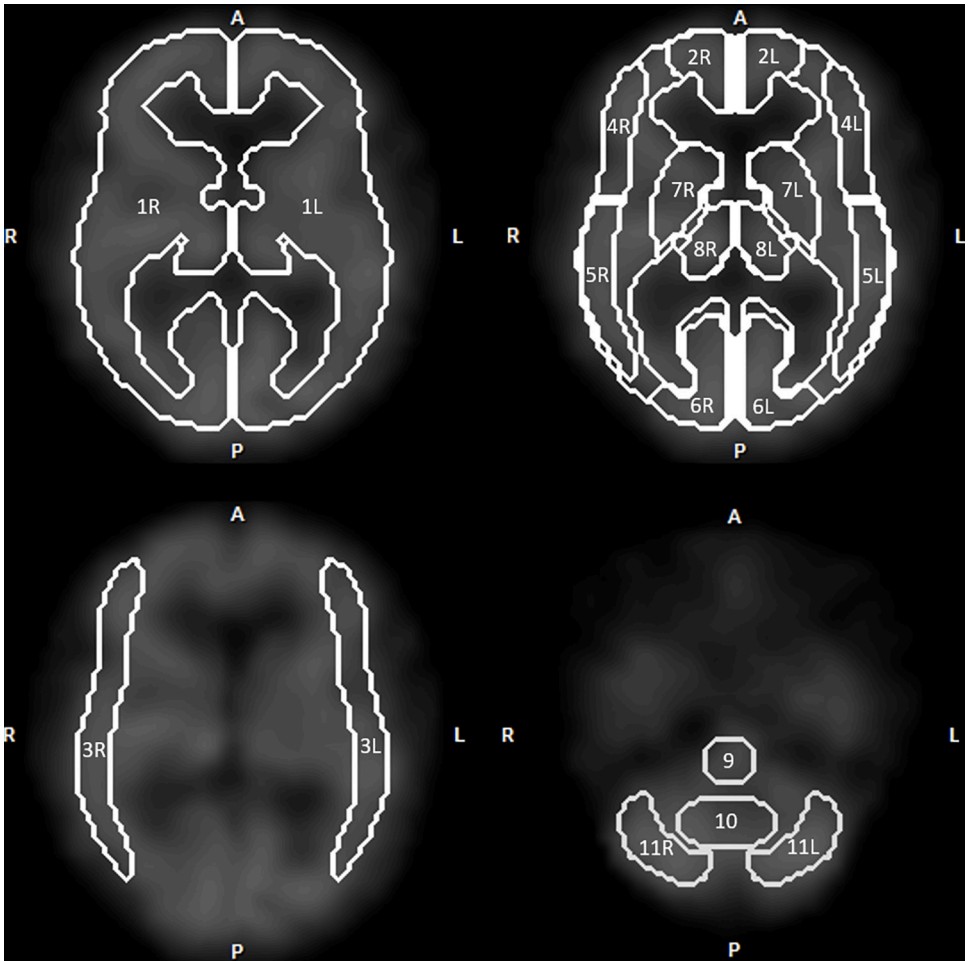

**Fig 3. The regions of interest identified by the NEURO FLEXER software.** Note. R, right; L, left; A, anterior; P, posterior; 1, hemisphere; 2, anterior cerebral artery; 3, middle cerebral artery (MCA); 4, anterior part of MCA; 5, posterior part of MCA; 6, posterior cerebral artery; 7, basal ganglia; 8, thalamus; 9, pons; 10, vermis; 11, cerebellum.

### Artificial neural network model

The dataset of the 294 participants included in the study was randomized into training ($n = 235$) and testing ($n = 59$) sets. The neural network module of MATLAB 2019a (Math-Works, Natick, MA, USA) was used to build the feedforward ANN regression model. In this study, the Bayesian optimization method was performed to automatically optimize the hyper-parameters of an ANN model, including the number of neurons in the hidden layer and activation functions in a network [14]. This ANN model was trained using the Levenberg–Marquardt backpropagation algorithm as a loss function [15], and a cross-validation was performed using the shuffle-split cross-validation method for estimating the accuracy of the model and to avoid overfitting [16]. Bayesian regularization was also performed to prevent overfitting [17]. The number of hidden layers and Epoch were manually set to 3 and 30, respectively.

### Estimation of the Ca10

The estimated Ca10 of the testing set was obtained through two ways. The first involved our proposed ANN model, whereas the second focused on the conventional calculation method using commercially available software [9].

## Calculation of rCBF and CVR

Using the estimated Ca10 as an input function, rCBFs at rest and after the acetazolamide challenge were calculated. CVR was calculated using the following equation:

CVR = (rCBF after acetazolamide challenge–rCBF at rest) × 100 / rCBF at rest

## Statistical analysis

Statistical analyses were conducted using MATLAB R2019a. The accuracy of our proposed ANN model and conventional model was evaluated using Pearson's correlation coefficient (r-value) between the measured and estimated Ca10, rCBF, and CVR. An r-value less than 0.2 was considered as poor, 0.2–0.5 as weak, 0.5–0.8 as moderate, and 0.8–1 as strong correlation. The Bland–Altman analysis [18] was conducted to demonstrate the agreement between the measured and estimated Ca10. In the Bland–Altman analysis, the 95% limits of agreement (LoA) were defined as the mean difference ± 1.96 × standard deviation.

## Results

The hyperparameters of our proposed feedforward ANN model were determined as an input layer with 28 items, 3 hidden layers with 3, 32, and 9 nodes, and an output layer (Fig 4). Log-sigmoid transfer function was used as activation functions between hidden layer 1 and 2, hyperbolic tangent function between hidden layer 2 and 3, and log-sigmoid transfer function between hidden layer 3 and output layer (Fig 4).

The measured and estimated values of the Ca10 in the testing set are summarized in Table 1. Fig 5 presents the correlation of Ca10 between the measured and estimated values. In our proposed ANN model, Pearson's r-value was 0.81 [0.70–0.88] ($P < .001$) (Fig 5A), which indicated strong positive correlation between the measured and estimated values. Contrarily,

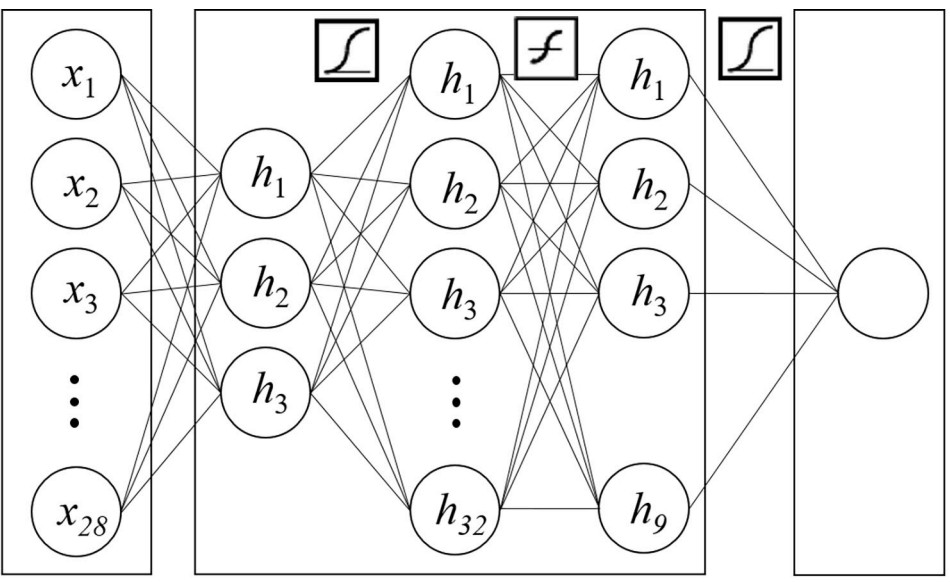

**Fig 4. Our proposed artificial neural network model consisted of a 28-item input layer and 3 hidden layers with 3, 32, and 9 nodes, respectively.** The activation functions defined the log-sigmoid transfer function between hidden layers 1 and 2, the hyperbolic tangent function between hidden layers 2 and 3, and the log-sigmoid transfer function between hidden layer 3 and the output layer.

**Table 1. The measured and estimated values of the Ca10 in testing set.**

| Test ID | Measured Ca10 | Estimated Ca10 | | Height (cm) | Weight (kg) | Background disease | Post treatment for cerebral circulation |
|---|---|---|---|---|---|---|---|
| | | Proposed model | Conventional method | | | | |
| 1 | 135.6 | 127.2 | 108.8 | 163 | 64 | Moyamoya disease | ◯ |
| 2 | 119.1 | 99.3 | 89.7 | 155 | 74 | Moyamoya disease | ◯ |
| 3 | 83.7 | 84.2 | 67.5 | 165 | 102 | Rt-ICA stenosis | |
| 4 | 120.6 | 143.4 | 117.3 | 140 | 61 | Rt-ICA occlusion, Lt-MCA occlusion | |
| 5 | 125.2 | 125.4 | 112.2 | 161 | 61 | Moyamoya disease | ◯ |
| 6 | 138.5 | 153.9 | 173.4 | 145 | 43 | Rt-ICA stenosis, Lt-ICA occlusion | |
| 7 | 100.1 | 102.5 | 94.2 | 154 | 73 | Rt-MCA stenosis | |
| 8 | 109.1 | 107.6 | 100.8 | 174 | 72 | Rt-ICA stenosis | |
| 9 | 128.3 | 135.5 | 151.0 | 164 | 50 | Lt-ICA occlusion | ◯ |
| 10 | 135.4 | 131.6 | 117.1 | 166 | 60 | Lt-MCA stenosis | |
| 11 | 73.5 | 81.4 | 77.4 | 175 | 90 | Lt-ICA stenosis | |
| 12 | 103.1 | 104.6 | 86.3 | 160 | 81 | Rt-ICA occlusion, Lt-ICA stenosis | ◯ |
| 13 | 73.5 | 93.2 | 99.8 | 170 | 60 | Rt-MCA occlusion | ◯ |
| 14 | 134.7 | 150.0 | 204.0 | 153 | 37 | Rt-ICA stenosis | |
| 15 | 107.4 | 101.7 | 99.4 | 160 | 70 | Rt-MCA occlusion | |
| 16 | 112.0 | 107.0 | 110.8 | 164 | 66 | Moyamoya disease | ◯ |
| 17 | 127.9 | 131.1 | 153.8 | 154 | 46 | Rt-ICA stenosis | |
| 18 | 119.8 | 123.4 | 122.0 | 159 | 62 | Lt-MCA occlusion | |
| 19 | 108.5 | 114.7 | 108.2 | 171 | 69 | Lt-MCA stenosis | |
| 20 | 143.0 | 140.0 | 142.5 | 154 | 48 | Moyamoya disease | ◯ |
| 21 | 100.9 | 115.6 | 112.9 | 168 | 61 | Lt-ICA stenosis | |
| 22 | 113.5 | 121.2 | 110.4 | 162 | 69 | Rt-ICA stenosis | |
| 23 | 100.9 | 104.3 | 116.9 | 163 | 65 | Rt-ICA stenosis | ◯ |
| 24 | 125.1 | 123.4 | 140.3 | 168 | 55 | Bilateral ICA stenosis | |
| 25 | 109.6 | 126.0 | 140.7 | 165 | 53 | Rt-ICA occlusion | |
| 26 | 67.6 | 88.2 | 80.2 | 181 | 97 | Rt-ICA occlusion | |
| 27 | 132.6 | 136.7 | 142.4 | 163 | 54 | Rt-MCA occlusion | ◯ |
| 28 | 83.0 | 94.2 | 103.0 | 163 | 67 | Rt-ICA stenosis | ◯ |
| 29 | 91.8 | 119.5 | 101.8 | 166 | 70 | Bilateral ICA stenosis | |
| 30 | 142.6 | 154.8 | 132.2 | 155 | 56 | Lt-ICA stenosis | |
| 31 | 99.4 | 122.5 | 128.1 | 170 | 60 | Rt-ICA stenosis | ◯ |
| 32 | 149.2 | 123.8 | 103.3 | 161 | 75 | Rt-ICA occlusion | |
| 33 | 119.4 | 121.3 | 120.3 | 160 | 61 | Lt-ICA stenosis | |
| 34 | 117.9 | 132.6 | 139.7 | 148 | 55 | Bilateral ICA stenosis | |
| 35 | 109.2 | 128.2 | 122.2 | 161 | 63 | Moyamoya disease | |
| 36 | 113.9 | 116.9 | 109.6 | 161 | 68 | Bilateral ICA stenosis | |
| 37 | 128.8 | 141.4 | 137.0 | 155 | 51 | Lt-ICA stenosis | |
| 38 | 100.7 | 115.2 | 97.9 | 169 | 72 | Rt-ICA stenosis | |
| 39 | 116.2 | 113.1 | 139.2 | 164 | 54 | Rt-ICA occlusion | |
| 40 | 121.2 | 126.7 | 116.0 | 163 | 56 | Bilateral SA stenosis | |
| 41 | 159.4 | 152.6 | 198.3 | 142 | 37 | Rt-MCA stenosis | |
| 42 | 119.0 | 143.7 | 151.6 | 150 | 50 | BA stenosis | |
| 43 | 114.6 | 123.7 | 121.0 | 155 | 56 | Rt-ICA stenosis, Lt-ICA occlusion | ◯ |
| 44 | 118.6 | 115.8 | 112.2 | 165 | 65 | Bilateral MCA stenosis | |
| 45 | 131.5 | 127.9 | 140.9 | 163 | 53 | Rt-ICA stenosis | |

(*Continued*)

**Table 1.** (Continued)

| 46 | 128.9 | 136.1 | 124.4 | 160 | 60 | Lt-MCA stenosis | |
|----|-------|-------|-------|-----|----|------------------|----|
| 47 | 102.8 | 97.9 | 93.2 | 186 | 80 | Lt-MCA stenosis | |
| 48 | 124.6 | 122.4 | 127.5 | 153 | 58 | Bilateral MCA stenosis, Aortic dissection | |
| 49 | 104.1 | 111.4 | 101.1 | 167 | 73 | Rt-CCA stenosis, Lt-ICA stenosis | |
| 50 | 100.3 | 115.6 | 129.2 | 162 | 58 | Lt-MCA occlusion | |
| 51 | 112.9 | 102.7 | 101.8 | 169 | 69 | Moyamoya disease | |
| 52 | 153.5 | 135.1 | 132.4 | 162 | 57 | dAVF | ◯ |
| 53 | 144.7 | 124.4 | 109.8 | 176 | 68 | Rt-VA stenosis | |
| 54 | 103.1 | 97.4 | 101.9 | 163 | 69 | Lt-ICA occlusion | |
| 55 | 112.1 | 127.6 | 104.6 | 150 | 64 | Lt-MCA occlusion | ◯ |
| 56 | 115.9 | 124.6 | 117.9 | 173 | 61 | Lt-VA stenosis | |
| 57 | 141.2 | 149.4 | 164.3 | 148 | 45 | Lt-ICA stenosis | |
| 58 | 103.7 | 109.7 | 113.8 | 168 | 64 | Rt-ICA stenosis, Lt-CCA stenosis | |
| 59 | 120.6 | 130.1 | 122.6 | 145 | 60 | Lt-MCA stenosis | |

Note. Ca10, 123I-IMP arterial blood radioactivity concentration; M, male; F, female; Rt, right; Lt. left; ICA, internal carotid artery; MCA, middle cerebral artery; SA, subclavian artery; BA, basilar artery; CCA, common carotid artery; dAVF, dual arteriovenous fistula; VA, vertebral artery.

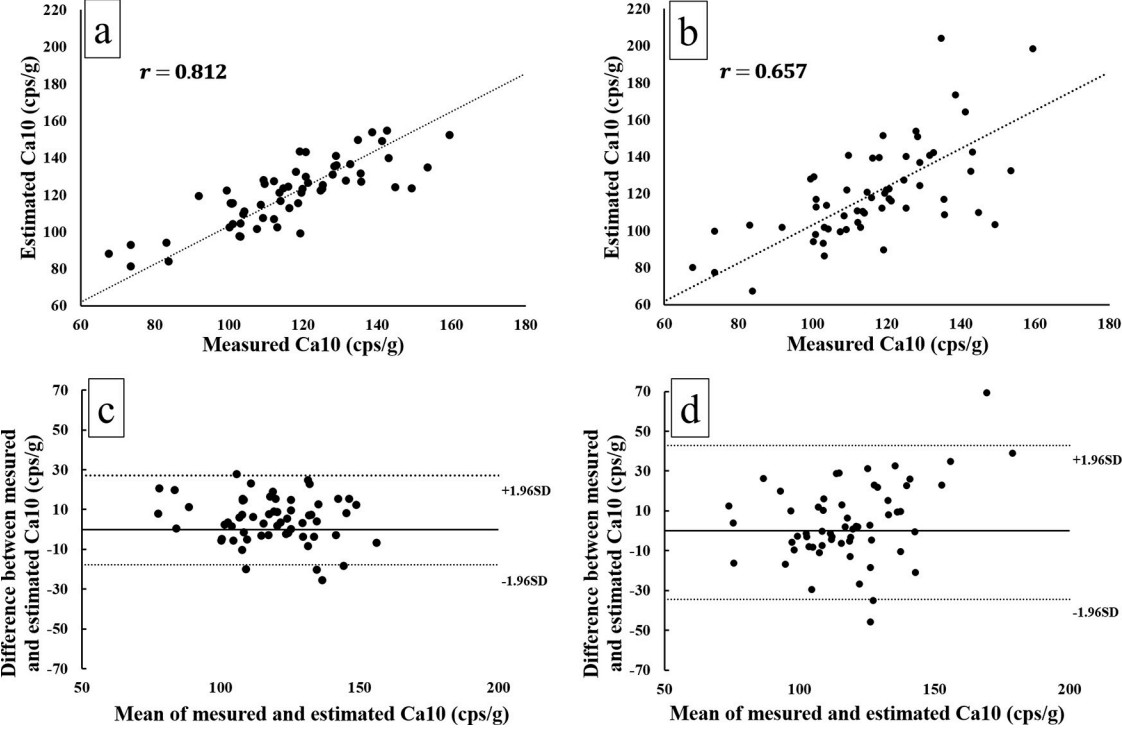

**Fig 5.** (a) Scatterplot with regression line between the true measured and estimated Ca10 by our proposed model revealed Pearson's r-value of 0.81. (b) Bland–Altman plots of the differences versus the mean values between the measured and estimated Ca10 revealed that the short-dashed lines denote the 95% limits of agreement, and these values had a small and balanced dissemination. (c) Scatterplot with regression line between the true measured and estimated Ca10 by conventional method revealed Pearson's r-value of 0.66. (d) Bland–Altman plots of the differences versus the mean values between the measured and estimated Ca10 revealed that these values had a relatively large and scattered dissemination.

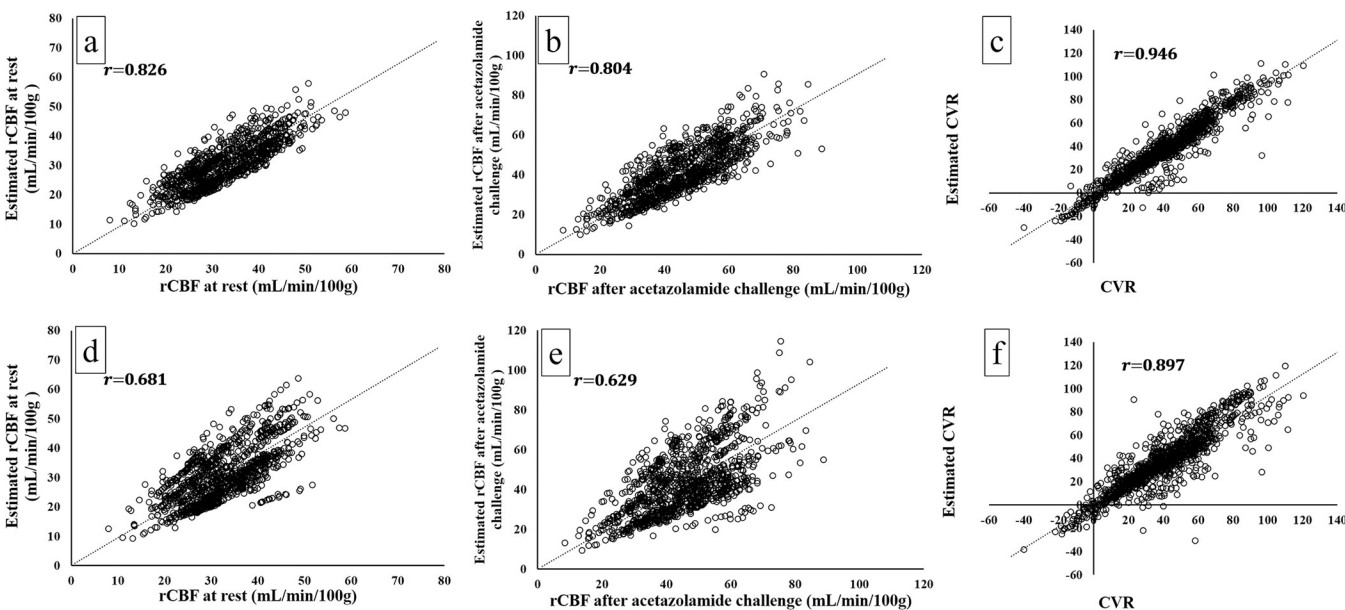

**Fig 6. Scatterplot with regression line between the true measured and estimated rCBF at rest (a) by our proposed model revealed Pearson's r-value of 0.83 and (d) by conventional method revealed Pearson's r-value of 0.68.** Scatterplot with regression line between the true measured and estimated rCBF after acetazolamide challenge (b) by our proposed model revealed Pearson's r-value of 0.80 and (e) by conventional method revealed Pearson's r-value of 0.63. Scatterplot with regression line between the true measured and estimated CVR (c) by our proposed model revealed Pearson's r-value of 0.95 and (f) by conventional method revealed Pearson's r-value of 0.90.

Pearson's r-value using the conventional method was 0.66 [0.48–0.78] ($P < .001$) (Fig 5B), indicating a moderate positive correlation between the measured and estimated values. Fig 5C presents the Bland–Altman plots that compare the measured and estimated Ca10 using our proposed ANN model. These values were found to have a small and balanced dissemination of relative differences and a significant fixed error ($P < .01$) with a mean difference of 4.7 (95% LoA: −18–27). Fig 5D presents the Bland–Altman plots that compare the measured and estimated Ca10 using the conventional method. These values were shown to have a relatively large and scattered dissemination of relative differences and a significant proportional error (slope of 0.355; $P < .01$) with a mean difference of 4.1 (95% LoA: −35–43).

Fig 6 presents the correlation of rCBF and CVR between the measured and estimated values. In our proposed ANN model, Pearson's r-value was 0.83 [0.81–0.84] ($P < .001$) in rCBF at rest (Fig 6A), 0.80 [0.78–0.82] ($P < .001$) in rCBF after the acetazolamide challenge (Fig 6B), and 0.95 [0.94–0.95] ($P < .001$) in CVR (Fig 6C), which indicated a strong positive correlation between the measured and estimated values. In the conventional method, Pearson's r-value was 0.68 [0.65–0.71] ($P < .001$) in rCBF at rest (Fig 6D), 0.63 [0.59–0.66] ($P < .001$) in rCBF after the acetazolamide challenge (Fig 6E), and 0.90 [0.89–0.91] ($P < .001$) in CVR (Fig 6F), which indicated a moderate or strong positive correlation between the measured and estimated values.

## Discussion

Using our feedforward ANN model, the Ca10, rCBF, and CVR in DTARG could be estimated both accurately and non-invasively. Therefore, our proposed model would have a distinct advantage in reducing the burden on both patients and physicians.

Currently, the ARG and DTARG methods have been widely used for rCBF measurements using 123I-IMP. However, both methods require a one-point arterial blood sampling for the input function. Alternatively, accurate estimation of Ca10 in ARG and DTARG allows patients

and physicians to skip the prerequisite invasive arterial blood sampling. Therefore, we attempted to estimate Ca10 in our institution using abundant DTARG data obtained by the same device and the same software. Our proposed ANN model demonstrates a strong linear correlation of Ca10, rCBF, and CVR between the true measured value and the estimated value. These results suggest that our proposed model could be applied to clinical practice, whereas rCBF via this method should be verified for determining the clinical significance compared with that of positron emission tomography with $^{15}$O-labelled water.

The present study provided a potential non-invasive estimation of the Ca10 using a feedforward ANN model. Our results indicated that there is a relationship among the Ca10, patient physique, total injection 123I-IMP radiation dose, and distribution of 123I-IMP count. Therefore, ML algorithms can resolve this complex and possibly nonlinear relationship. In our proposed ANN model, optimization for the hyperparameters of an ANN was carefully considered by introducing the Bayesian optimization method, and overfitting was avoided by employing Bayesian regularization and shuffle-split cross-validation. Consequently, an appropriate learning model could be deployed. In our proposed ANN model, Pearson's r-value was 0.81, it could not be denied that the accuracy was insufficient. However, it is worth noting that the methodology of the prediction model using ANN was proven to be feasible enough for refinement. Additionally, our model has scope for further improvement. The number of Epoch empirically set to 30 might lead to overfitting and underlearning. It could be expected that the accuracy would be improved by increasing the hidden layer that we set to three, i.e., adapting deep learning. In addition, our method can improve the data settings. More specifically, data from all 294 participants who underwent rCBF measurements through the DTARG method with 123I-IMP were included in this study to create the ANN model. However, almost all 294 participants had abnormal cerebral blood flow due to preexisting conditions. Therefore, provided that the ANN model was trained and tested separately for various patterns, such as normal cerebral blood flow, abnormal cerebral blood flow, and localization of abnormal cerebral blood flow with underlying diseases, the accuracy of estimation would be improved.

Certain non-invasive methods for rCBF quantification had been developed but never became popular. For instance, the non-invasive microsphere method uses the cardiac output and pulmonary clearance of 123I-IMP for estimating the input function [19]. However, this method is not popular owing to its complicated procedures, such as measurement of the cardiac output on the day of inspection using the ultrasonic Doppler method. Abe *et al.* have established a similar non-invasive estimation method for Ca10 using dynamic acquisition data from the lungs and brain [20]. This method is relatively simple but requires additional acquisition of dynamic data from the chest. Alternatively, our proposed ANN model does not require any procedural changes or installation of additional equipment for the measurement of rCBF.

The present study has several limitations. First, the study was conducted at a single center, which could have caused a selection bias. In addition, our proposed ANN model is device- and software-dependent, and thus, it cannot be implemented in other institutions without modifications. Contrarily, we believe that it is feasible to build a similar ANN model customized to other specific devices in a similar manner or perform transfer learning using our trained model. Second, with regard to rCBF and CVR, further investigation is essential in order to establish clinical significance.

## Conclusions

Our proposed neural network-based algorithm could estimate the Ca10, rCBF, and CVR accurately and non-invasively in DTARG. This proposed method may help to reduce the burden on both patients and physicians.

## Supporting information

**S1 Table. The raw datasets of the 294 participants including participants' demographics, SPECT data, and the Ca10.** Note. M, male; F, female; Ca10, 123I-IMP arterial blood radioactivity concentration; CCF, cross-calibration factor; ACA, anterior cerebral artery; MCA, middle cerebral artery; M2ant, anterior part of MCA; M2post, posterior part of MCA; PCA, posterior cerebral artery; BG, basal ganglia.
(XLSX)

## Acknowledgments

The authors of this manuscript declare no relationships with any companies whose products or services may be related to the subject matter of the article.

## Author Contributions

**Conceptualization:** Tetsuro Kaga, Hiroki Kato, Toyohiro Imai, Tomohiro Ando, Yoshifumi Noda, Takayuki Miura, Yukiko Enomoto, Fuminori Hyodo, Toru Iwama, Masayuki Matsuo.

**Data curation:** Tetsuro Kaga, Toyohiro Imai, Tomohiro Ando, Takayuki Miura, Yukiko Enomoto.

**Formal analysis:** Tetsuro Kaga, Toyohiro Imai, Tomohiro Ando, Yoshifumi Noda.

**Investigation:** Tetsuro Kaga, Hiroki Kato, Toyohiro Imai, Tomohiro Ando, Yukiko Enomoto, Fuminori Hyodo, Masayuki Matsuo.

**Methodology:** Tetsuro Kaga, Hiroki Kato, Toyohiro Imai, Yoshifumi Noda, Takayuki Miura, Fuminori Hyodo.

**Project administration:** Tetsuro Kaga.

**Software:** Tetsuro Kaga, Toyohiro Imai, Takayuki Miura.

**Supervision:** Hiroki Kato, Yoshifumi Noda, Toru Iwama, Masayuki Matsuo.

**Validation:** Hiroki Kato, Toyohiro Imai, Tomohiro Ando, Yoshifumi Noda, Yukiko Enomoto.

**Visualization:** Tetsuro Kaga, Toyohiro Imai.

**Writing – original draft:** Tetsuro Kaga.

**Writing – review & editing:** Hiroki Kato, Fuminori Hyodo, Toru Iwama, Masayuki Matsuo.

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
