## [Decision Letter · Decision Letter 0]

18 Nov 2022

PONE-D-22-10899Non-invasive regional cerebral blood flow quantification in the 123I-IMP autoradiography using artificial neural networkPLOS ONE

Dear Dr. Kaga,

Thank you for submitting your manuscript to PLOS ONE. After careful consideration, we feel that it has merit but does not fully meet PLOS ONE’s publication criteria as it currently stands. Therefore, we invite you to submit a revised version of the manuscript that addresses the points raised during the review process.

We look forward to receiving your revised manuscript.

Kind regards,

Anwar P.P. Abdul Majeed

Academic Editor

PLOS ONE

https://journals.plos.org/plosone/s/fileid=ba62/PLOSOne_formatting_sample_title_authors_affiliations.pdf.

Additional Editor Comments:

1. Please kindly explicitly mention the hyperparameters used for the ANN mode in-text. I did notice it mentioned in the figure, kindly move the information to the methods section.

2. There is a discrepancy in the following statement :

"In this study, the Bayesian optimization method was performed to automatically optimize the hyperparameters of an ANN model, including the number of hidden layers, number of neurons in the hidden layer, and activation functions in a network [14]."

However a few sentence later the author claimed the following : "The number of hidden layers and Epoch were manually set to 3 and 30, respectively."

Kindly confirm if it was selected or was it optimised?

Reviewers' comments:

Reviewer's Responses to Questions

**Comments to the Author**

1. Is the manuscript technically sound, and do the data support the conclusions?

Reviewer #1: Yes

2. Has the statistical analysis been performed appropriately and rigorously? 

Reviewer #1: Yes

3. Have the authors made all data underlying the findings in their manuscript fully available?

Reviewer #1: Yes

4. Is the manuscript presented in an intelligible fashion and written in standard English?

Reviewer #1: Yes

5. Review Comments to the Author

Reviewer #1: Using data from 294 participants who underwent DTRAG in the author's institution, this retrospective study exploits an artificial neural network based-method using machine learning that does not require arterial blood sampling for estimating Ca10, rCBF, CVR in DTRAG.

Scope for improvements are there through further investigations. Besides eliminating arterial blood sampling this method could also be useful for improving our understanding about the the altered rCBF and associated cognitive deficiencies in patients undergoing chemotherapy, substance abuse, PTSD etc.

6. PLOS authors have the option to publish the peer review history of their article (what does this mean?). If published, this will include your full peer review and any attached files.

Reviewer #1: No

---

## [Author Response · Author response to Decision Letter 0]

21 Nov 2022

We thank the reviewer very much for the kind and expert review. We revised our manuscript along the comments.

For Reviewer #1’s comments:

1. Please kindly explicitly mention the hyperparameters used for the ANN mode in-text. I did notice it mentioned in the figure, kindly move the information to the methods section.

- Thank you for your comment. We described the hyperparameters used for the ANN in the first paragraph of methods section. We kindly ask for your confirmation.

2. There is a discrepancy in the following statement:

"In this study, the Bayesian optimization method was performed to automatically optimize the hyperparameters of an ANN model, including the number of hidden layers, number of neurons in the hidden layer, and activation functions in a network [14]. However a few sentence later the author claimed the following: "The number of hidden layers and Epoch were manually set to 3 and 30, respectively. "Kindly confirm if it was selected or was it optimized?

- We sincerely apologize for the inconvenience caused by our mistake. As you mentioned, the discrepancy occurred about decision procedure for the number of hidden layers. In the present study, the number of hidden layers was manually determined, not automatically optimized by Bayesian optimization method. Therefore, we deleted the phrase "the number of hidden layers" from the sentence about the Bayesian optimization method. We kindly ask for your confirmation.

-----

Additionally, we revised our manuscript along with PLOS ONE's style requirements and described the full name of IRB.

---

## [Decision Letter · Decision Letter 1]

16 Jan 2023

PONE-D-22-10899R1

Non-invasive regional cerebral blood flow quantification in the 123I-IMP autoradiography using artificial neural network

PLOS ONE

Dear Dr. Kaga,

Thank you for submitting your manuscript to PLOS ONE. After careful consideration, we feel that it has merit but does not fully meet PLOS ONE’s publication criteria as it currently stands. Therefore, we invite you to submit a revised version of the manuscript that addresses the points raised during the review process.

We look forward to receiving your revised manuscript.

Kind regards,

Anwar P.P. Abdul Majeed

Academic Editor

PLOS ONE

Journal Requirements:

Additional Editor Comments:

Thank you for addressing the question - although your reply was inaccurate, however the changes in the manuscript was reflected accurately.

Please also check the following.

Page 9 - Fig6s or Fig6a?

Reviewers' comments:

Reviewer's Responses to Questions

**Comments to the Author**

1. If the authors have adequately addressed your comments raised in a previous round of review and you feel that this manuscript is now acceptable for publication, you may indicate that here to bypass the “Comments to the Author” section, enter your conflict of interest statement in the “Confidential to Editor” section, and submit your "Accept" recommendation.

Reviewer #1: All comments have been addressed

2. Is the manuscript technically sound, and do the data support the conclusions?

Reviewer #1: Yes

3. Has the statistical analysis been performed appropriately and rigorously? 

Reviewer #1: Yes

4. Have the authors made all data underlying the findings in their manuscript fully available?

Reviewer #1: Yes

5. Is the manuscript presented in an intelligible fashion and written in standard English?

Reviewer #1: Yes

6. Review Comments to the Author

Reviewer #1: Thanks for making changes and revising the manuscript in alignment with the comments made by the reviewers.

7. PLOS authors have the option to publish the peer review history of their article (what does this mean?). If published, this will include your full peer review and any attached files.

Reviewer #1: No

---

## [Author Response · Author response to Decision Letter 1]

20 Jan 2023

We thank the reviewer and editor very much for the kind and expert review. We revised our manuscript along the comments.

For Editor‘s comments:

Thank you for addressing the question - although your reply was inaccurate, however the changes in the manuscript was reflected accurately.　Please also check the following.

Page 9 - Fig6s or Fig6a?

- Thank you for your comment. The description “Fig 6s” in Page 9 line 16 was correct. We described “Fig 6s” as including Fig 6a to 6f, but it could be confusing. We changed the description “Fig 6s” to simply “Fig 6”. For the same reason, the description “Fig 5s” in Page 9 line 2 was changed to “Fig 5”. If you have any other opinion or intention, please let me know.

In addition, we added supporting information S1 including raw data sets used for machine learning.

---

## [Editor Report · Decision Letter 2]

5 Feb 2023

Non-invasive regional cerebral blood flow quantification in the 123I-IMP autoradiography using artificial neural network

PONE-D-22-10899R2

Dear Dr. Kaga,

We’re pleased to inform you that your manuscript has been judged scientifically suitable for publication and will be formally accepted for publication once it meets all outstanding technical requirements.

Kind regards,

Anwar P.P. Abdul Majeed

Academic Editor

PLOS ONE
---

## [Editor Report · Acceptance letter]

27 Feb 2023

PONE-D-22-10899R2 

Non-invasive regional cerebral blood flow quantification in the 123I-IMP autoradiography using artificial neural network 

Dear Dr. Kaga:

I'm pleased to inform you that your manuscript has been deemed suitable for publication in PLOS ONE. Congratulations! Your manuscript is now with our production department. 

Kind regards, 

on behalf of

Dr. Anwar P.P. Abdul Majeed 

Academic Editor

PLOS ONE